# Does Control-Related Information Attenuate Biased Self-Control and Moral Perceptions Based on Weight?

**DOI:** 10.3390/bs15070970

**Published:** 2025-07-17

**Authors:** Casey L. Timbs, Heather M. Maranges

**Affiliations:** Department of Psychology, Florida State University, Tallahassee, FL 32304, USA; maranges@psy.fsu.edu

**Keywords:** self-control, moral perceptions, body weight, intervention

## Abstract

Negative weight-based attitudes are pervasive and difficult to change. One reason may be the moralization of weight: if people use higher weight as a cue for lower self-control, they may infer lower moral character, given the strong link between self-control and morality. Moralized attitudes tend to be resistant to change. Accordingly, we tested whether (1) people perceived others with higher (vs. lower) weight as having lower self-control and, in turn, morality and (2) whether targeting control-related perceptions attenuated the weight → self-control → morality links. To that end, in two preregistered experiments (see OSF), we employed intervention strategies targeting control-related perceptions to increase moral evaluations of higher-weight individuals. Specifically, we provided evidence of a higher-weight person’s (a) weight uncontrollability (Study 1) and (b) high self-control (Study 2). People perceived higher-weight targets as having lower self-control, and this predicted perceptions of lower moral character. However, as with extant weight-based attitude interventions, neither experimental intervention strategy attenuated less positive (i.e., made more positive) moral character perceptions. These findings suggest that it is not enough to intervene on control-related beliefs to reduce the moralization of weight. We suggest intervening on moral perceptions directly and the possibility that moralization of weight may be automatic, requiring interventions targeting automatic attitudes.

## 1. Introduction

Body weight-based discrimination is one of the last “socially acceptable” forms of bias ([30]; [52]), with both implicit and explicit forms continuing to increase or hold steady while other types of bias decrease ([9], [10]). Negative attitudes toward people with higher weight are largely driven by beliefs that weight is controllable ([11]; [28]; [67]), and, therefore, higher weight reflects a lack of self-control ([11]; [54]; [56]). Efforts to intervene on weight bias have often been unsuccessful or yielded small effect sizes (see [42], for a meta-analysis and review). One reason for the pervasive nature of weight-based bias may be that the attitude is in part moralized ([35]; [56]). The moralization of weight is unsurprising given that people perceive weight to be predictive of self-control and that self-control is essential to morality (e.g., [19]). Indeed, [35] ([35]) show that people view others with higher versus lower weight as lower in self-control and, in turn, morality. Moral perceptions and judgments are particularly difficult to intervene on given moralized attitudes tend to be deeply held, strong, and difficult to change ([34]; [56]). Accordingly, the current work aims to intervene on the less positive moral perceptions of people with higher versus lower weight by providing countervailing information about the target’s (a) weight uncontrollability and (b) high self-control in two preregistered studies.

Weight bias is a pervasive negative attitude toward people with higher weight, held explicitly and implicitly, across varied age groups ([49]) and cultures ([12]; [37]). Some research has suggested that weight bias is one of the only remaining acceptable and openly expressed forms of bias and is perceived as less negative than other forms of bias ([9], [10]; [30]; [52]). Weight bias plays out in diverse domains such as healthcare ([52]; [53]; [67]; [68]), education ([52]; [53]), parenting ([52]; [53]), the workplace ([8]; [44]; [53]), and close relationships ([52]; [53]). The emergent weight-based discrimination can then create highly consequential inequities as well as negative effects on health and well-being ([26], [25]; [39]; [64]; [71]), especially among women, who are judged more harshly for having higher weight ([18]; [27]; [49]).

A growing body of work attempts but largely fails to intervene on anti-fat bias, suggesting that anti-fat attitudes are particularly robust and resistant to change (e.g., [20]; [29]; [69]). Weight bias may be hard to intervene on due to the self-control-related content of the bias and the subsequent moralization of weight ([35]; [56]). Indeed, the content of the weight bias is consistently negatively valenced (e.g., incompetent, unintelligent, unsociable versus competent, intelligent, successful at relationships; [6]; [23]; [31]) but is predominated by negative stereotypes that focus on characteristics related to low self-control ([6]; [38]; [54]; [72], [73]). For example, people attribute lower willpower ([6]), self-discipline, conscientiousness, and emotional stability ([72]) and higher laziness ([6]; [56]; [72]) to people with higher versus lower weight. 

Low self-control attributions seem to arise because people believe that weight is controllable and that people with higher weights must therefore lack self-control ([11]; [28]; [67]). For example, people who recently lost weight, regardless of current body weight, judged others with higher weight more harshly; crucially, the more negative judgments were due to stronger beliefs that weight is controllable ([3]). Although most interventions on weight bias are unsuccessful, mitigating the perceptions of controllability of weight can somewhat attenuate anti-fat bias, and that attitude change is accounted for by reductions in beliefs that weight is under individuals’ control ([5]; [14]; [45]). For instance, people assigned to an intervention course in which they studied, discussed, and presented scientific research indicating that weight is often *uncontrollable* (e.g., biologically predisposed) showed reduced implicit and explicit anti-fat attitudes compared to conditions in which people engaged in the same activities but focused on the *controllability* of weight (e.g., caused by overeating) or something unrelated (e.g., alcohol; [45]).

People attend to signals of others’ self-control because the ability is essential for cooperation, prosocial behaviors, and overriding selfish and antisocial impulses that may undermine the goals or well-being of the group—self-control predicts moral character (e.g., [2]; [13]; [19]; [74]). Indeed, people infer moral character from success in self-control, even in non-moral domains ([22]; [41]), which reflects the real relationship between morality and self-control. For example, per ecological momentary assessment, people with stronger moral views about a behavioral choice (i.e., not eating meat) better self-regulate and overcome temptations (i.e., to eat meat; [7]). However, when cues of self-control are not overt, people rely on stereotypic, though inconsistent or inaccurate, cues of self-control, such as body weight, to inform person perceptions ([11]; [72]). Thus, people may hold a self-control-focused anti-fat bias that promotes a moral-character-focused anti-fat bias. Indeed, recent work finds that people view those with higher weight as lower in self-control, and therefore lower in morality ([35]). Moralized attitudes tend to be strongly entrenched and hard to change ([34]). Accordingly, the primary goal of the current work is to experimentally test whether countervailing, explicit information on (a) the uncontrollability of weight and (b) the high self-control of a higher-weight person can attenuate perceptions of the higher-weight person as having low self-control and low moral character.

### The Current Work

People use body weight as a cue of self-control, and this bias contributes to a moralization of higher weight ([35]). Although it may be difficult to shift moralized weight-based bias because moral attitudes are particularly resistant to change, it may be possible to shift moral perceptions by shifting proximally predictive self-control perceptions. Via two preregistered studies, the current work tests whether providing information that a higher-weight target’s weight is out of their control (Study 1) or providing evidence that the higher-weight target has high self-control (Study 2) can weaken perceptions of low self-control and therefore boost perceptions of moral character. Specifically, in Study 1 we provided information that a target had either a biological disorder that caused weight gain (i.e., weight is uncontrollable) or a biological disorder similarly named but unrelated to weight. We expected that the *uncontrollability-of-weight*, versus *unrelated-to-weight*, condition would weaken the association between higher weight and lower self-control and therefore moral character. Specifically, we expected that perceptions of self-control and, in turn, moral character, would be higher in the uncontrollability-of-weight condition compared to the unrelated-to-weight condition. Study 2 entailed a 2 × 2 design in which we provided information that a target (either higher-weight or lower-weight) had either *high self-control* or *no information about self-control*, with similarly positive information across conditions. We expected that providing information about a target’s high self-control would weaken the tendency to rely on weight as a cue of self-control and therefore moral character. In both studies, we first tested whether we could weaken the self-control perceptions associated with weight, and second, whether targeting the self-control perceptions would lead to more positive moral character perceptions of the higher-weight individuals.

We controlled for participants’ BMI and subjective weight. We also addressed the halo effect, that is, that positive and negative evaluative content across traits and domains tends to covary; people who are seen as beautiful are also seen as good and smart, and so on, whereas people who are seen as ugly are also seen as bad and unintelligent, and so on ([15]; [55]; [75]; [76]). Given that both self-control and morality are viewed as positive, we wanted to ensure that results were not due to an overall halo effect. We therefore controlled for perceptions of the targets’ attractiveness, disgustingness (Studies 1 and 2), competence, power, and warmth (Study 2) because these perceptions tend to covary with each other, as well as perceptions of weight (e.g., [60]; [63]; [70]; [72]), and morality (e.g., [56]; [62]).

Notably, we focus on women as evaluative targets for a few reasons. First, women, versus men, tend to be judged more negatively for having higher weight (e.g., [18]; [59]). Second, research linking weight and self-control has primarily focused on women (e.g., [33]; [48]). Third, men and women are viewed as differing in levels of various moral traits (e.g., [32]; [43]), making comparisons across mixed-gender targets difficult. For preregistration and data, see the Open Science Framework (OSF).

## 2. Study 1

Study 1 entailed an experimental design in which participants were randomly assigned to “meet” a higher-weight target who either had a biological disorder that made her gain weight uncontrollably via metabolic dysregulation, or a biological disorder that sounded similar but was unrelated to her weight via melanin dysregulation. The *uncontrollability-of-weight*, versus *unrelated-to-weight*, information was expected to weaken the association between higher weight and low self-control and higher weight and low moral character, as well as the indirect effect of condition on moral perceptions through self-control perceptions. Put another way, we expected that participants in the uncontrollability-of-weight condition would perceive the target as having higher self-control and therefore higher moral character than those in the unrelated-to-weight condition.

On the other hand, much research suggests that moral and person perception is deep-seated and largely automatic (e.g., [16]), and self-protective stereotyping is largely outside the purview of explicit processing (e.g., [4]; [36]). To the extent that the link between people’s perceptions of a higher-weight target’s self-control and moral character rely on automatic or intuitive rather than explicit or deliberative processing, people’s biased perceptions about the moral character of a target may be unchanged by explicit information that their weight is out of their control. Study 1 thus tested these competing hypotheses.

### 2.1. Method

***Participants.*** An a priori power analysis for a predicted effect size of 0.26 for X→M and M→X mediation effects indicated that 162 people would be needed to reach 80% power ([21]). We increased our power by collecting data from 245 people via MTurk who opted to participate in a study on first impressions. We decided a priori to exclude people who failed an attention check or to recall information about the target (i.e., rated the target as “underweight” and did not recall any information about the target’s disorder in an open response question; *n* = 34), leaving a final sample of 211 people (*M*_age_ = 39.96, *SD* = 13.34; 110 women, 101 men; 172 White, 19 Black, 16 Asian, 3 Native American or Alaska Native, 1 Native Hawaiian or Other Pacific Islander, of whom 11 also identified as Hispanic or Latino/a). The study took approximately 20 min, and participants were compensated $1.00 (USD).

### 2.2. Procedure and Materials

After providing consent, participants were randomly assigned to either the *uncontrollable-weight-disorder* condition or the *unrelated-to-weight-disorder* condition. All participants “met” a woman via picture and short introduction. Specifically, in both conditions, they viewed a photograph of the same woman with higher weight and read the target’s purported introduction. The introductions were identical except one described a disorder that causes weight gain (i.e., a metabolic disorder) whereas the other described a disorder that is unrelated to weight (i.e., a melanin disorder). See Appendix A and Appendix B for photographs and introductions, respectively.

***Self-control perceptions.*** Participants rated the target’s self-control using an adapted version of the 13-item Brief Self-Control Scale ([66]), using a response scale from 1 (*strongly agree*) to 7 (*strongly disagree*). Sample items included the following: *This person is good at resisting temptation* and *This person wishes she had more self-discipline* (reversed; averaged across conditions: *M* = 4.29, *SD* = 0.88, α = 0.89). See Appendix C for the full scale.

***Moral character perceptions.*** Participants responded to 27 items on a scale from 1 (*strongly disagree*) to 7 *(strongly agree*; averaged across conditions: *M* = 5.16, *SD* = 0.85, α = 0.96). As examples, participants responded to items such as *This person… is basically honest, is trustworthy, is moral, is a good person, and does not have morals* (reversed; [77]). For the full scale, see Appendix D.

***Control measures.*** Participants used a 1 (*strongly disagree*) to 7 (*strongly agree*) scale to rate the target on attractiveness (i.e., *This person is attractive*; averaged across conditions: M = 2.27, SD = 1.34) and disgust (i.e., *This person is disgusting*; averaged across conditions: *M* = 3.50, *SD* = 1.54), among distractor items (see Appendix E). Participants then answered questions about their own weight and height (1 = *extremely underweight* to 7 = *extremely overweight*, *M* = 4.70, *SD* = 1.03; BMI, *M* = 27.38, *SD* = 6.93) as well as manipulation check questions about the target (Appendix F). 

### 2.3. Results and Discussion

First, we tested whether people perceived the higher-weight target with an *uncontrollable weight disorder* as having more self-control than the higher-weight target with an *unrelated disorder* via a two-tailed independent samples *t*-test. There was no significant difference in self-control ratings of the target with the uncontrollable weight disorder (*M* = 4.36, *SD* = 0.84) compared to the target with the unrelated disorder (*M* = 4.23, *SD* = 0.93), *t*(211) = −1.10, *p =* 0.273, *d* = −0.15. See Figure 1 and Table 1 for descriptives.

Second, we tested whether people perceived the higher-weight target with an *uncontrollable weight disorder* as having stronger moral character than the higher-weight target with an *unrelated disorder* via a two-tailed independent samples *t*-test. There was no significant difference in moral character perceptions of the target with the uncontrollable weight disorder (*M* = 5.17, *SD* = 0.78) compared to the target with the unrelated disorder (*M* = 5.15, *SD* = 0.92), *t*(211) = −0.21, *p =* 0.832, *d* = 0.02), as displayed in Figure 1 and Table 1. See Appendix G for control analyses.
Figure 1Perceptions of self-control and moral character across conditions, Study 1. **Note.** Error bars represent +/− 1 standard error of the mean.
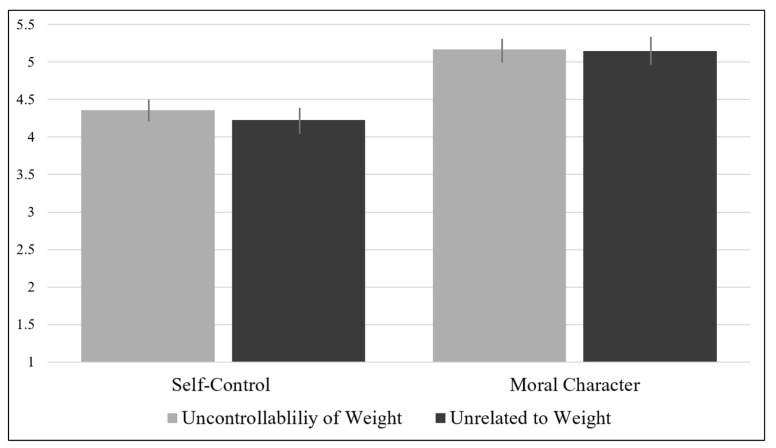

behavsci-15-00970-t001_Table 1Table 1Perceptions of self-control and moral character, Study 1.
Self-ControlMoral CharacterUncontrollable WeightUnrelated to WeightUncontrollable WeightUnrelated to WeightMean4.364.235.175.15SD0.840.930.780.92


We also employed Bayesian independent *t*-tests to better assess the extent to which we could accept versus reject the null hypotheses (H_0_)—that there were no differences in perceptions of self-control or morality of a higher-weight individual based on whether she had a weight-related disorder that caused weight gain or a weight-unrelated disorder—relative to predicted differences by condition (H_1_). Comparing the null (0) to the alternative hypothesis (1), a Bayes factor >1 provides evidence that the null is likely true compared to the alternative hypothesis, whereas Bayes factor <1 provides evidence of the opposite, that the alternative is likely true compared to the null.

First, we compared perceptions of self-control across conditions, using a diffuse prior without the assumption of equal variance and employing [57]’s ([57]) method of Bayes factor computation (such that results are consistent with the default of other statistical programs, such as R). Results yielded a negligible difference between the posterior means of the target with the weight-related disorder (*M* = 4.23, *SD* = 0.93) and the target with the weight-unrelated disorder (*M* = 4.36, *SD* = 0.84), with a 95% credible interval of [−0.108, 0.376] and a Bayes factor favoring the null hypothesis, BF_01_ = 5.16. Consistent with the frequentist results, we could not reject the null; moreover, we had strong evidence to accept the null with respect to perceptions of self-control. Second, we conducted the same analysis with the outcome of perceptions of morality comparing posterior means across the weight-related disorder (*M* = 5.17, *SD* = 0.78) and the weight-unrelated disorder (*M* = 5.15, *SD* = 0.92) conditions, finding strong evidence to accept the null: a 95% credible interval of [−0.209, 0.259] and BF_01_ = 9.06.

Although there were no significant differences in self-control perceptions or moral character perceptions of the two targets, perceptions of self-control were associated with perceptions of morality across targets, *r* = 0.50, *p* < 0.001.

Finally, as preregistered, we tested whether perceptions of self-control would carry significant indirect variance from condition (uncontrollable weight disorder or unrelated disorder) to perceptions of target moral character via 10,000 bootstrapping resample mediation analyses using Model 4 in the PROCESS Macro for SPSS Version 29. The indirect effect of self-control perceptions was not significant, *b* = 0.06, *SE* = 0.06, 95% CI [−0.046, 0.198]; see Figure 2. We replicated this pattern controlling for self-reported BMI and target’s perceived attractiveness and disgustingness (see Appendix G).

Study 1 assessed whether people’s tendency to use higher weight as a cue of low self-control and, in turn, moral character was sensitive to the controllability of an individual’s weight. Our results suggest not, that is, we were not able to intervene on self-control- or morality-based anti-fat bias by providing countervailing, explicit information that a higher-weight person’s weight is uncontrollable due to a metabolic disorder. Although self-control ratings were positively associated with moral character ratings, people perceived the same higher-weight woman as similar in self-control and morality regardless of whether she had a metabolic glandular disorder that caused her high weight. Perhaps moralized weight bias will not shift without overt evidence of a higher-weight individual’s having high self-control. Put another way, intervening on beliefs about self-control itself may be particularly important—the aim of the next study.

## 3. Study 2

Study 2 tested whether explicit individualizing and countervailing information about an individual’s *high* self-control would attenuate reliance on weight as a cue of self-control and therefore moral character. This study also improved upon Study 1 by varying weight. Specifically, we employed a 2 (weight: higher or lower) × 2 (self-control information: high or none) design. We expected that self-control information would moderate (weaken) the association between weight and self-control perceptions, which in turn would be associated with moral perceptions of the target. In other words, it may be that people will perceive the higher-weight target with evidence of high self-control as being higher in self-control and moral character than the higher-weight target with *no* evidence related to self-control. We compared perceptions across the four resulting targets, tentatively expecting that the lower-weight target with evidence of high self-control would be viewed as having the highest self-control and moral character, followed by the higher-weight target with evidence of high self-control, with the lowest ratings for the two targets, both higher- and lower- weight, with no evidence of high self-control. Of course, these predictions relied on the assumption that dependence on weight as a cue of self-control and therefore morality is under the purview of explicit, deliberative processes. If these moral perceptual patterns are more underpinned by automatic versus deliberative processes (e.g., [16]), then providing countervailing, individuating information about a novel other’s self-control may not lead to differential evaluations of two people with higher weight that differ in information about their self-control. Indeed, [61] ([61]) suggest that the current design (i.e., a moderation of process design) is informative to the extent that the proposed psychological process is a conscious one.

### 3.1. Method

***Participants***. An a priori G*power analysis for a predicted interaction effect size of *f* = 0.25 (a medium effect) suggested that 287 people would be needed to reach 90% power ([17]). In order to increase our power and due to expected attention and memory check failures, we decided a priori to collect data from at least 400 people via MTurk. Four hundred and twenty-two people completed the survey. We applied strict inclusion criteria, excluding people who failed one or both of the two attention checks or to respond accurately to memory questions about the target’s weight and informational paragraph (*n* = 111), leaving a final sample of 311 (133 women, 176 men, 1 genderfluid, 1 nonbinary; *M*_age_ = 35.18, *SD* = 9.87; participants could select multiple race/ethnicities: 234 White, 33 Black, 25 Asian, 13 Hispanic/Latinx, 3 Native American or Alaska Native, 2 Native Hawaiian or Other Pacific Islander, 1 Middle Eastern). The study took approximately 20 min, and participants were compensated $1.00 USD.

### 3.2. Procedure and Materials

Participants were randomly assigned to “meet” one of four targets, per the 2 (weight: higher or lower) × 2 (self-control information: high or none) design, by seeing her photograph and reading her introductory paragraph. The photo was of one woman either before (higher-weight) or after (lower-weight) weight loss (weight condition) and the introductory paragraph included information diagnostic (e.g., won an award at work for high self-discipline, carefully plans financial savings) or nondiagnostic (e.g., met Tom Hanks at the airport, watches favorite TV shows) of high self-control (self-control information condition). All conditions included similar information about the target (e.g., spends her free time with family and friends) and were positively framed across conditions to avoid valence’s driving effects. See Appendix A and Appendix H for all photographs and introductions, respectively.

Participants then rated the target on self-control (on 1–5 scale, *M* = 3.64, *SD* = 0.82, α = 0.92) and moral character (on 1–7 scale, *M* = 5.51, *SD* = 0.83, α = 0.96), as in Study 1. In addition to the same control items of disgust (*M* = 2.24, *SD* = 1.53) and attractiveness (*M* = 4.44, *SD* = 1.73), Study 2 also controlled for power (i.e., *This person is powerful*; *M* = 4.16, *SD* = 1.36), competence (i.e., *This person is competent*; *M* = 5.52, *SD* = 1.23), and warmth (i.e., *This person is warm*; *M* = 5.47, *SD* = 1.10) among distractor items. See Appendix E for all distractor items. Participants then answered questions about their own demographics and weight (1 = *extremely underweight* to 7 = *extremely overweight*, *M* = 4.41, *SD* = 1.05; BMI, *M* = 25.25, *SD* = 6.28) as well as the target’s weight and informational paragraph (i.e., manipulation and attention checks, Appendix F). 

### 3.3. Results

First, we tested whether our manipulation of self-control worked. It did: as expected, participants in the *high self-control information* condition (*M* = 3.95, *SD* = 0.81) viewed the target as having higher self-control than did those in the *no self-control information* condition (*M* = 3.32, *SD* = 0.70), *t*(309) = −7.41, *p* < 0.001, *d* = 0.76 (see Table 2 and Table 3 for descriptive statistics across all conditions).

Second, in our critical analysis, we tested whether countervailing information about a higher-weight person’s *high self-control* (vs. *no self-control information*) weakened the association between target weight and perception of target’s moral character. To do this, we conducted a 2 (target weight: higher or lower) × 2 (target self-control information: high or none) ANOVA with perceptions of moral character as the outcome. To the extent that self-control-related weight bias feeds into moralized weight bias via explicit, conscious processes, we should expect an interaction, such that the higher-weight target evidencing high self-control is viewed as having stronger moral character than the higher-weight target with no diagnostic self-control information. However, there was no significant main effect of weight condition, self-control information condition, nor interaction. See Figure 3 for the means and Table 4 for the ANOVA output. We replicated this pattern by controlling for self-reported BMI and target’s perceived attractiveness, disgustingness, power, competence, and warmth (see Appendix G).

As in Study 1, we employed Bayesian analyses. In conducting Bayesian ANOVAs (again, with Rouder’s computation and parameters described in Study 1), we compared the alternative hypothesis (H_1_) to the null one (H_0_, or the intercept). Accordingly, here, comparing the alternative hypothesis (1) to the null (0), a Bayes factor >1 provides evidence that the alternative hypothesis is likely true over the null hypothesis, whereas BF < 1 provides evidence of the opposite. First, we assessed the effect of self-control information (high or none) and weight (higher vs. lower) conditions on perceptions of self-control. There were differences across condition combinations in perceptions of self-control (see Appendix I, Table A4), with posterior *M* = 0.51, a 95% credible interval of [0.434, 0.596], and BF_10_ = 1.97 × 10^16^. That is, there was extremely strong evidence that the alternative hypothesis—that perceptions of self-control vary by self-control information × weight conditions—was true over the null hypothesis—that self-control perceptions do not vary by those conditions. Next, we assessed whether there were differences across condition combinations in perceptions of moral character. Converging with the frequentist results, there were no differences across condition combinations in perceptions of moral character (see Appendix I, Table A5), with posterior *M* = 0.69, a 95% credible interval of [0.585, 0.804], and BF_10_ = 0.001. This provided strong evidence that the alternative hypothesis should be rejected, and the null hypothesis should be accepted.

Importantly, again, across targets, perceptions of self-control were associated with perceptions of morality, *r* = 0.56, *p* < 0.001. This suggests that people are using perceptions of self-control to make inferences about moral character but not systematically based on self-control information.

We next conducted an exploratory moderated mediation analysis in which we tested whether self-control information condition (none = 0 vs. high = 1) moderated the link between weight condition (lower = 0 vs. higher = 1) and perceptions of self-control, which in turn were associated with perceptions of moral character. To do so, we employed the PROCESS macro Model 7 with 10,000 bootstrapped resamples. As expected, weight condition was significantly associated with perceptions of self-control, and perceptions of self-control were associated with perceptions of moral character (Figure 4). Put another way, people viewed the higher- versus lower-weight target as having lower self-control, which cued lower moral character perceptions. Self-control information did not moderate that pattern, with *b* = 0.08, *SE* = 0.16, and 95% CI [−0.242, 0.392], given that the indirect effect of weight condition on moral perceptions through self-control perceptions was significant for both the *no self-control information* condition, with *b* = −0.35, *SE* = 0.07, 95% CI [−0.508, −0.217], and the *high self-control information* condition, with *b* = −0.31, *SE* = 0.08, 95% CI [−0.469, −0.155]. Corroborating prior analyses, mediation results suggested that people were using weight as a cue of self-control to inform moral evaluations regardless of countervailing and relevant information. That is, again, we were not able to intervene on moralized weight bias by providing information about control, here about a target’s high self-control, even when changing self-control perceptions. 

## 4. General Discussion

Past work has found that people view those with higher weight as low in self-control, and therefore low in moral character ([35]). That pattern helps make sense of why weight-based biases are difficult to intervene on: they likely have a moral component, making the attitude deeply held and hard to change ([34]; [56]). The current work aimed to test whether we could shift weight-based self-control perceptions, and therefore moral evaluations, by providing information that the target’s weight was out of their control (Study 1) and by providing information that the higher-weight target had high self-control (Study 2).

Much like other attempts to intervene on weight bias, these experiments did not provide positive support for potential scalable interventions. Neither explicit countervailing information about the source of an individual’s higher weight, namely, a metabolic disorder that caused weight gain that was out of her control (Study 1) nor about her having high self-control (Study 2) shifted people’s reliance on weight as a cue of self-control and therefore moral character. The Bayesian analyses confirmed that we should accept the null hypotheses, that such explicit information, even when it shifted self-control perceptions, did not systematically shift moral character perceptions. This hints at the possibility that people hold automatic associations between higher (vs. lower) weight and lower (vs. higher) self-control and morality. Indeed, [35] ([35]) found that people held an automatic association between weight and self-control and weight and moral character via the IAT, and these two associations were positively correlated. This means that use of weight as a cue of self-control and morality may be more automatic, and that the more people view weight as a signal of self-control, the more they view weight as a signal of moral character.

Together with [35]’s ([35]) work, these results put into question the theoretical model that links *explicit* perceptions of self-control and morality. Shifting weight-based explicit perceptions of self-control, which predict explicit perceptions of moral character, did not shift explicit perceptions of moral character. With the insight that the weight–self-control and the weight–morality implicit attitudes are associated, one possible update is that the perceptions of the weight → self-control → moral character model is best understood as functioning *automatically.* Both deliberate and automatic attitudes contribute to explicit summary attitudes ([46]), such that intervening with explicit, countervailing information shifts only one contributor to the self-control–morality link, presumably with the automatic associations continuing to drive the summary explicit perceptual evaluations. Next, we discuss the implications of this updated perspective.

Other studies have detailed the difficulty of successfully intervening on weight bias and contend that more work is needed to understand why (see [29]; [42]; for reviews). Our work is consistent with work demonstrating that interventions aimed at reducing weight bias are unsuccessful for biases that emerge when measured with implicit association tests (IATs; e.g., [40]; [58]; [65]; [69]). Recent reviews have found that there has been limited success in weakening automatic anti-fat bias, including by strong interventions ([45]; see [29]; for review). Other work has found that implicit weight bias has only been “reduced” successfully by creating negative evaluative associations between lower-weight and negativity by using evaluative conditioning. For example, [29] ([29]) found that participants scored lower in automatic anti-fat attitudes after actively pairing higher-weight targets with positive stimuli and lower-weight targets with negative stimuli. Reducing positivity/increasing negativity toward one group to bring attitudes closer to that of another group seems on its face ethically unacceptable and fails to truly intervene on the problem. Our work combines with others’ to suggest that intervening on automatic attitudes tied to self-control and morality, perhaps via evaluative conditioning, could serve as a promising next step given that explicit interventions have not been successful.

It cannot be understated how important it is to consider the content of stereotypes in order to effectively intervene on them. For example, other work has found that it is not a mere negativity association that underpins anti-Black bias, rather it is a threat association ([36]). Attitudes that entail an association with threat may make that bias especially pervasive. Similarly, attitudes tied to morality tend to be closely held, strong, and difficult to change ([34]; [56]). For bias to be reduced, the intervention must match the content of the association. This work and that of [35] ([35]) make it clear that the association between weight and self-control likely engenders associations between weight and moral character. It is not merely negative. As mentioned above, interventions may include self-control and morality-related evaluative conditioning. But they may need to go beyond that. Decades of research on the intergroup contact hypothesis ([1]) has demonstrated that under certain conditions, negative biases can be reduced via interactions with the members of the stereotyped group (for meta-analyses, see [47]; and [50]; for systematic review, see [51]). Those conditions include participants having equal status in the contact context, common goals, cooperation, and the support of authorities, law, or customs ([1]). However, if those conditions are not met and the interaction is negative, the intervention can not only fail, but it can also increase negative attitudes. With this in mind, we suggest that future work test the efficacy of intergroup contact interventions that both create opportunities for people of differing weights to interact and facilitate evidencing of self-control and moral character, making the cooperation context criterion all the more important.

The current studies have many strengths, including novel theorizing, experimental designs, appropriate power, and preregistration, but there are limitations worth noting. First, albeit intentional, the use of women targets precludes the generalizability of results to non-women targets. Second, the targets were white and young middle-aged women. Thus, future research should aim to replicate the studies presented here with targets of more varied gender, age, and race/ethnicity to extend the generalizability of the results. Relatedly, participants were in the United States and lived in a WEIRD society (Western, Educated, Industrialized, Rich, Democratic; [24]). Other countries and cultures may hold different views about weight, self-control, and morality, such that this work should be conducted elsewhere and with non-American participants. Third, although Study 1 included a manipulation check of self-control, future studies should include a direct manipulation check of the uncontrollability of the target weight. Finally, future research may benefit from alternative operationalizations or forms of these experiments. For example, it may benefit the intervention for participants to meet people in person (vs. in pictures), and self-control and moral character perceptions may be measured via automatic measures (e.g., IATs).

In conclusion, the current work tested whether providing information on the uncontrollability of weight (Study 1) as well as countervailing information that a higher-weight target has high self-control (Study 2) could successfully shift the perceptions of lower self-control and therefore moral character of those with higher-weight. Contrary to our hypotheses, neither intervention strategy resulted in significant, systematic improvement moral character evaluations. Although providing information that the target had high self-control did increase self-control perceptions, this did not affect moral evaluations of the target. Our work underscores the existence of self-control- and moral-character-based negative attitudes toward people with higher weights and adds to a growing body of research suggesting that negative weight-based evaluations may be difficult to intervene on explicitly.

## Figures and Tables

**Figure 2 behavsci-15-00970-f002:**
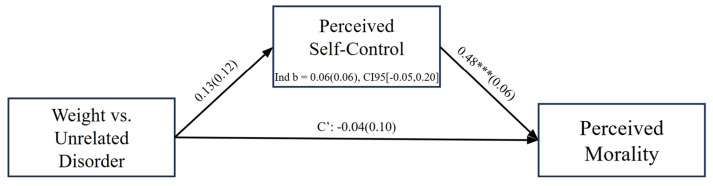
Mediation of the link between disorder condition (weight-related vs. -unrelated) and perceptions of moral character through self-control, Study 1. **Note.** *** *p* < 0.001. Condition coded as unrelated = 0 vs. weight-related disorder = 1.

**Figure 3 behavsci-15-00970-f003:**
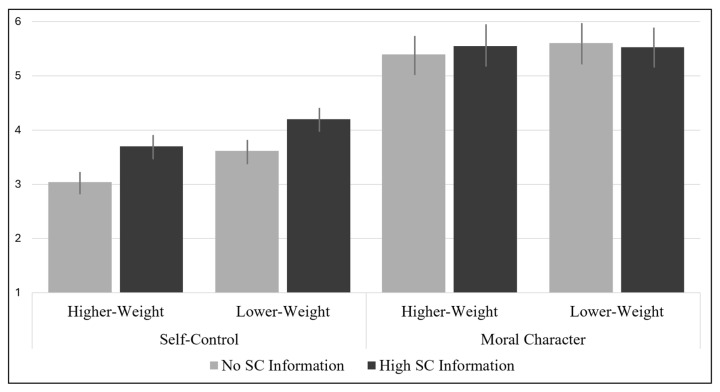
Perceptions of self-control and moral character across conditions, Study 2. **Note.** Self-control was assessed on a 5-point scale; moral character was assessed on a 7-point scale. Error bars represent +/− 1 standard error of the mean. SC = self-control.

**Figure 4 behavsci-15-00970-f004:**
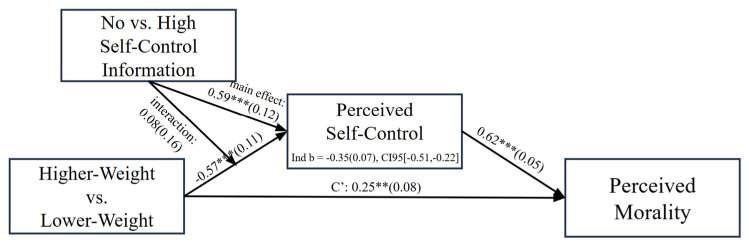
Moderated mediation with self-control information moderating the link between weight condition and perceptions of self-control, which in turn is associated with perceptions of moral character, Study 2. **Note**. *** *p* < 0.001, ** *p* < 0.01. Self-control information condition coded as none = 0 vs. high = 1. Weight condition coded as lower = 0 vs. higher = 1.

**Table 2 behavsci-15-00970-t002:** Perceptions of self-control by condition, Study 2.

	Higher Weight		Lower Weight	
	No SC Information	High SC Information	No SC Information	High SC Information
Mean	3.042	3.703	3.615	4.202
SD	0.558	0.794	0.729	0.748

**Note.** SC = self-control.

**Table 3 behavsci-15-00970-t003:** Perceptions of moral character by condition, Study 2.

	Higher Weight		Lower Weight	
	No SC Information	High SC Information	No SC Information	High SC Information
Mean	5.393	5.547	5.603	5.525
SD	0.691	0.953	0.856	0.786

**Note.** SC = self-control.

**Table 4 behavsci-15-00970-t004:** Output for the 2 (weight: higher or lower) × 2 (self-control information: high or none) ANOVA predicting perceptions of moral character, Study 2.

Predictor	F	*p*
Intercept	13,872.073	<0.001
Weight Condition	0.997	0.319
Self-Control Information Condition	0.168	0.682
Weight Cond × SC Info Cond	1.541	0.215

## Data Availability

The original data presented in the study are openly available on the OSF at https://osf.io/chuvx/?view_only=d33bc327763a4844b6a2a749a414886f (accessed on 2 April 2025).

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
