# Peer review of "Does Control-Related Information Attenuate Biased Self-Control and Moral Perceptions Based on Weight?"

_behavsci, 2025, doi:10.3390/bs15070970_

Round 1
Reviewer 1 Report
Comments and Suggestions for Authors
The present work examined whether experimental manipulations can decrease the impact of self-control on moralized person perceptions in the domain of weight attributions. Study 1 did so by manipulating perceived controllability (the mediator variable) and failed to observe the target effect. Study 2 provided the same results using an even stronger test by directly manipulating perceived self-control. These results are surprising and useful for future research in multiple ways, and the paper is well-written. The intro provides interesting information and a convincing rationale for the research questions. The authors do a great job in communicating the arguments in a simple but informative and convincing way. Nonetheless, I want to take the opportunity of this review to raise a couple of questions and suggestions for minor revision.
It is always difficult to experimentally manipulate control perceptions about real-life topics. However, the sample sizes reported in this paper provide sufficient statistical power, and together with the statistics the results reflect not only merely nonsignificant effects. Instead, based on the statistics and sample sizes, I guess the results are most likely informative, i.e. they show evidence of the absence of the hypothesized effects. If the authors want to consider making this argument, they could run Bayesian analyses or equivalence tests to probe the absence of the hypothesized tests and thereby increase the informativeness of the results even more. That being said, I expect readers will draw the same kind of strong conclusions from the reported statistics as mine, largely regardless of such additional analyses.
The authors’ interpretation of the findings highlights ways for slightly refining the conceptual model. That makes sense. Perhaps it might also be worth considering to discuss whether the observed findings might have far-reaching implications for the suggested causality of the model’s association – In my opinion, the elephant in the room is whether the model’s assumptions about the causal associations need to be reconsidered altogether? I am not sure and would be eager to hear the authors’ opinion on this important question.
In Study 2, please report the means and SD of the manipulation checks and key variables by all levels of interaction of the experimental manipulations.
Given that the concepts of self-control and moralization are central to the paper and interrelated, this recent paper might be relevant to consider:
Buttlar, B., Pauer, S., Scherrer, V., & Hofmann, W. (2024). Attitude-Based Self-Regulation: An Experience Sampling Study on the Role of Attitudes in the Experience and Resolution of Self-Control Conflicts in the Context of Vegetarians. Motivation Science. https://doi.org/10.31234/osf.io/mvzrn
Lastly, the abstract could be slightly improved to more accurately reflect the paper, especially by highlighting its interventional aspects, as it would be a pity if interested readers miss out on this aspect. And the following could be rephrased to further increase clarity: “we tested interventions on negative weight-based evaluations by providing evidence of (a) the uncontrollability of a target’s weight (Study 1) and (b) a target’s high self-control”; “neither experiment 18 attenuated less positive moral character perceptions”.
Author Response
Reviewer 1:
The present work examined whether experimental manipulations can decrease the impact of self-control on moralized person perceptions in the domain of weight attributions. Study 1 did so by manipulating perceived controllability (the mediator variable) and failed to observe the target effect. Study 2 provided the same results using an even stronger test by directly manipulating perceived self-control. These results are surprising and useful for future research in multiple ways, and the paper is well-written. The intro provides interesting information and a convincing rationale for the research questions. The authors do a great job in communicating the arguments in a simple but informative and convincing way. Nonetheless, I want to take the opportunity of this review to raise a couple of questions and suggestions for minor revision. 
Thank you for your close read and constructive feedback. We have sought to address your concerns in a systematic way.
- It is always difficult to experimentally manipulate control perceptions about real-life topics. However, the sample sizes reported in this paper provide sufficient statistical power, and together with the statistics the results reflect not only merely nonsignificant effects. Instead, based on the statistics and sample sizes, I guess the results are most likely informative, i.e. they show evidence of the absence of the hypothesized effects. If the authors want to consider making this argument, they could run Bayesian analyses or equivalence tests to probe the absence of the hypothesized tests and thereby increase the informativeness of the results even more. That being said, I expect readers will draw the same kind of strong conclusions from the reported statistics as mine, largely regardless of such additional analyses.
This is a great idea. We now include Bayesian analyses in both studies and discuss the theoretical implication of having strong evidence to accept the null in many cases. See pages 6 and 9 for results and 12 for discussion.
- The authors’ interpretation of the findings highlights ways for slightly refining the conceptual model. That makes sense. Perhaps it might also be worth considering to discuss whether the observed findings might have far-reaching implications for the suggested causality of the model’s association – In my opinion, the elephant in the room is whether the model’s assumptions about the causal associations need to be reconsidered altogether? I am not sure and would be eager to hear the authors’ opinion on this important question.
This is a great question. For what it is worth, we show this mediation model works, suggesting the manipulation doesn’t work but the links among the model do. This suggests that the links in the model are robust to updating (Maranges et al., 2025). One alternative model is of the halo effect, that people view others with higher weight more negatively across trait domains, and we are just capturing the covariance among them. However, the pattern is robust to things like disgust, attractiveness, power, competence, and warmth here and in Maranges et al. (2025). What might need to be updated is that this person-perceptual process is likely automatic. Accordingly, we discuss this possibility in the general discussion on page 12.
“With the insight that the weight–self-control and the weight–morality implicit attitudes are associated, one possible update is that the perceptions of weight à self-control à moral character model is best understood as functioning automatically. Both deliberate and automatic attitudes contribute to explicit summary attitude reports (Olson & Fazio, 2008), such that intervening with explicit, countervailing information shifts only one contributor to the self-control–morality link, presumably with the automatic associations continuing to drive the summary explicit perceptual evaluations.”
- In Study 2, please report the means and SD of the manipulation checks and key variables by all levels of interaction of the experimental manipulations.
We have added these results to our paper (see Tables 1, 2, and 3).
- Given that the concepts of self-control and moralization are central to the paper and interrelated, this recent paper might be relevant to consider: Buttlar, B., Pauer, S., Scherrer, V., & Hofmann, W. (2024). Attitude-Based Self-Regulation: An Experience Sampling Study on the Role of Attitudes in the Experience and Resolution of Self-Control Conflicts in the Context of Vegetarians. Motivation Science. https://doi.org/10.31234/osf.io/mvzrn
Thank you for sharing this interesting paper. We have added it to our paper (page 3) and agree that it has important relevance to our background and theory.
“For example, per ecological momentary assessment, people with stronger moral views about a behavioral choice (i.e., not eating meat), better self-regulate and overcome temptations (i.e., to eat meat; Buttlar et al., 2024).”
- Lastly, the abstract could be slightly improved to more accurately reflect the paper, especially by highlighting its interventional aspects, as it would be a pity if interested readers miss out on this aspect. And the following could be rephrased to further increase clarity: “we tested interventions on negative weight-based evaluations by providing evidence of (a) the uncontrollability of a target’s weight (Study 1) and (b) a target’s high self-control”; “neither experiment attenuated less positive moral character perceptions”.
We have sought to increase the clarity of our abstract in terms of interventional approach and findings according to your feedback and that of Reviewer 2. Of course, if we misunderstood your feedback, we happy to update the language in more specific ways
“Abstract: Background. Negative weight-based attitudes are pervasive and difficult to change. One reason may be the moralization of weight: if people use higher-weight as a cue for lower self-control, they may infer lower moral character, given the strong link between self-control and morality. Moralized attitudes tend to be resistant to change. Accordingly, we test whether (1) people perceive others with higher- (vs. lower-) weight as lower in self-control and, in turn, morality and (2) targeting control-related perceptions attenuates the weightàself-controlàmorality links. Method. To that end, in two preregistered experiments (see OSF), we employed intervention strategies targeting control-related perceptions to increase moral evaluations of higher-weight individuals. Specifically, we provided evidence of a higher-weight person’s (a) weight uncontrollability (Study 1) and (b) high self-control. Results. People perceived higher-weight targets as having lower self-control, and this predicted perceptions of lower moral character. However, as with extant weight-based attitude interventions, neither experimental intervention strategy attenuated less positive (i.e., made more positive) moral character perceptions. Conclusion. These findings suggest that it is not enough to intervene on control-related beliefs to reduce the moralization of weight. We suggest intervening on moral perceptions directly and the possibility that moralization of weight may be automatic, requiring interventions targeting automatic attitudes.”
Thank you for all of your thoughtful feedback. Its application has greatly improved the manuscript!

Reviewer 2 Report
Comments and Suggestions for Authors
I am grateful for the opportunity to have reviewed the manuscript entitled, “Does Control-Related Information Attenuate Biased Self-Control and Moral Perceptions Based on Weight?” I found the paper to be thorough and interesting, and I think it makes a useful contribution to the literature on underpinnings of anti-fat bias. Below, I have enumerated some comments and suggestions that I have for strengthening the manuscript.
- First, I commend the authors’ commitment to pre-registering both studies. I had hoped to view their pre-registrations, but for some reason, the hyperlinks in the document did not work for me, so I was not able to. In a future round, could the full link be provided (i.e., rather than an embedded hyperlink), so that they are viewable?
- On page 2 (lines 59-61), I was not sure what was meant by “Weight bias can be understood as weight stigma when focusing on deleterious effects on the well-being of higher-weight individuals…”. Weight stigma is generally used as an umbrella term, encompassing both anti-fat bias and weight-based discrimination, so the distinction here left me a bit confused.
- I believe there is a typo on page 3 (line 140); “OSF” stands for “Open Science Framework,” rather than “Foundation.”
- The literature review is thorough and nuanced, and it convinced me that the approach taken in this work is novel despite a wide array of prior literature investigating weight controllability beliefs and weight stigma.
- At the end of the Introduction, the authors note that they are controlling for participants’ own BMI and self-reported weight as well as some of their perceptions of the target, but the justification for why these have been included is under-justified. Please explain why these covariates were selected/pre-registered.
- Similarly, the mention of the halo effect was a bit under-developed; please remove it or expand on how it relates to the selection of covariates.
- Relatedly, please clarify which models include these covariates. I imagine they were not included in the t-tests or ANOVA, but were included in the mediation models?
- Were the manipulations for Study 1/Study 2 pilot tested? I ask because I wonder if some of the details included unintentionally manipulated a bit more than just the intended constructs. For instance, the phrase “and famous doctors have studied it” in the Study 1 vignettes is a bit unusual, and for those in the uncontrollable weight condition, it may have come across as defensive (i.e., because people frequently experience weight stigma, and therefore might be vigilant or defensive in response to future stigma). Of course, the phrase was also present for the unrelated disorder condition, but because paleness is not necessarily as stigmatized as having a higher weight, the phrase may not have been perceived as defensive in the unrelated disorder condition. Thus, it is possible that the study manipulation went beyond manipulating the controllability of the target’s weight, and incidentally manipulated how defensive the target seemed, or else made participants doubt the veracity of the target’s statement. This may have accounted, in part, for the lack of effect seen. I understand that data has been collected and these details cannot be changed, but if any pilot testing was done, it would be helpful to report that, and if not, it is worth noting the lack of manipulation piloting or a direct manipulation check (of perceptions of the controllability of their partner’s weight) as limitations of Study 1.
- On a couple of occasions (i.e., in abstract, in discussion section on line 416), the study manipulations are described as “interventions,” which may overstate the scope of the work presented. Perhaps a more appropriate way to describe these studies would be tests of “intervention strategies.”
- Figure 1 and Figure 3 would be easier to interpret if outcomes were grouped side-by-side, to allow for direct visual comparisons of means. For instance, in Figure 1, the two grouped bars could be “Self-Control” and “Moral Character,” and the colors indicated study condition. Figure 2 would be more interpretable if the bars were grouped by outcome (e.g., all self-control on the left, all moral character on the right) and then the two IVs were marked by color and secondary position.
- The final sentence in the discussion, “Put another way, the current work highlights the novel implication that interventions on negative weight-based perceptions may be most effective if directed at the automatic level and associations with self-control and moral character” goes a bit beyond the data presented here. I appreciated the discussion of automatic-level approaches as a future direction, but I think it is out of place as the final sentence of the paper. The penultimate sentence is an adequate conclusion that better represents the studies presented.
Author Response
I am grateful for the opportunity to have reviewed the manuscript entitled, “Does Control-Related Information Attenuate Biased Self-Control and Moral Perceptions Based on Weight?” I found the paper to be thorough and interesting, and I think it makes a useful contribution to the literature on underpinnings of anti-fat bias. Below, I have enumerated some comments and suggestions that I have for strengthening the manuscript.
Thank you for your close read and constructive feedback. We have sought to address your concerns in a systematic way.
- First, I commend the authors’ commitment to pre-registering both studies. I had hoped to view their pre-registrations, but for some reason, the hyperlinks in the document did not work for me, so I was not able to. In a future round, could the full link be provided (i.e., rather than an embedded hyperlink), so that they are viewable?
Apologies. The full link is available at the end of the manuscript on page 14.
- On page 2 (lines 59-61), I was not sure what was meant by “Weight bias can be understood as weight stigma when focusing on deleterious effects on the well-being of higher-weight individuals…”. Weight stigma is generally used as an umbrella term, encompassing both anti-fat bias and weight-based discrimination, so the distinction here left me a bit confused.
Thank you for clarifying. We have removed language of weight stigma and focus on anti-fat bias, about which we theorize and do operationalize.
- I believe there is a typo on page 3 (line 140); “OSF” stands for “Open Science Framework,” rather than “Foundation.”
Thank you. We fixed this typo.
- The literature review is thorough and nuanced, and it convinced me that the approach taken in this work is novel despite a wide array of prior literature investigating weight controllability beliefs and weight stigma.
Thank you for this positive feedback. We are glad that an expert views the introduction as sufficiently comprehensive.
- At the end of the Introduction, the authors note that they are controlling for participants’ own BMI and self-reported weight as well as some of their perceptions of the target, but the justification for why these have been included is under-justified. Please explain why these covariates were selected/pre-registered.
We have added more cites as well as expanded on our justification for these in the introduction on page 4.
“We control for participants’ BMI and subjective weight. We also address the halo effect. That is, positive and negative evaluative content across traits and domains tends to covary; people who are seen as beautiful are also seen as good and smart, and so on, whereas people who are seen as ugly are also seen as bad and unintelligent, and so on (Eagly et al., 1991; Puhl et al., 2011; Wade & DiMaria, 2003; Wade et al., 2003). Given that both self-control and morality are viewed as positive, we want to ensure that results are not due to an overall halo effect. We therefore control for perceptions of the target’s attractiveness, disgustingness (Studies 1 and 2), competence, power, and warmth (Study 2) because these perceptions tend to covary with each other, perceptions of weight (e.g., Schwartz et al., 2006; Stewart & Ogden, 2018; Vartanian, 2010; Vartanian et al., 2013), and morality (e.g., Ringel & Ditto, 2019; Spielmann et al., 2024).”
- Similarly, the mention of the halo effect was a bit under-developed; please remove it or expand on how it relates to the selection of covariates.
Please see quoted text above, where we explain and expand on the halo effect.
- Relatedly, please clarify which models include these covariates. I imagine they were not included in the t-tests or ANOVA, but were included in the mediation models?
We added clarification to this for both Study 1 (page 6) and Study 2 (page 10). We replicated analyses for both studies.
- Were the manipulations for Study 1/Study 2 pilot tested? I ask because I wonder if some of the details included unintentionally manipulated a bit more than just the intended constructs. For instance, the phrase “and famous doctors have studied it” in the Study 1 vignettes is a bit unusual, and for those in the uncontrollable weight condition, it may have come across as defensive (i.e., because people frequently experience weight stigma, and therefore might be vigilant or defensive in response to future stigma). Of course, the phrase was also present for the unrelated disorder condition, but because paleness is not necessarily as stigmatized as having a higher weight, the phrase may not have been perceived as defensive in the unrelated disorder condition. Thus, it is possible that the study manipulation went beyond manipulating the controllability of the target’s weight, and incidentally manipulated how defensive the target seemed, or else made participants doubt the veracity of the target’s statement. This may have accounted, in part, for the lack of effect seen. I understand that data has been collected and these details cannot be changed, but if any pilot testing was done, it would be helpful to report that, and if not, it is worth noting the lack of manipulation piloting or a direct manipulation check (of perceptions of the controllability of their partner’s weight) as limitations of Study 1.
We did not formally do pilot testing. Informally, we workshopped the information together and sought to match them on features like positivity and uniqueness, conversing with research assistants about their impressions. Study 1 lacking a manipulation check other than self-control is a limitation, though study 2’s is self-control ratings. We added discussion of this in our limitations section (page 13).
- On a couple of occasions (i.e., in abstract, in discussion section on line 416), the study manipulations are described as “interventions,” which may overstate the scope of the work presented. Perhaps a more appropriate way to describe these studies would be tests of “intervention strategies.”
We changed “interventions” in reference to our work to “intervention strategies”.
- Figure 1 and Figure 3 would be easier to interpret if outcomes were grouped side-by-side, to allow for direct visual comparisons of means. For instance, in Figure 1, the two grouped bars could be “Self-Control” and “Moral Character,” and the colors indicated study condition. Figure 2 would be more interpretable if the bars were grouped by outcome (e.g., all self-control on the left, all moral character on the right) and then the two IVs were marked by color and secondary position.
These have now been changed to reflect the suggested groupings. Both figures now group self- control and moral character separately.
- The final sentence in the discussion, “Put another way, the current work highlights the novel implication that interventions on negative weight-based perceptions may be most effective if directed at the automatic level and associations with self-control and moral character” goes a bit beyond the data presented here. I appreciated the discussion of automatic-level approaches as a future direction, but I think it is out of place as the final sentence of the paper. The penultimate sentence is an adequate conclusion that better represents the studies presented.
Thank you for your input on this. We certainly do not want to speak beyond the data. We removed this sentence and ended on a statement which our data supports.
Thank you for your feedback! The manuscript is stronger for it.

Round 2
Reviewer 2 Report
Comments and Suggestions for Authors
Overall, I found the authors to be quite responsive to my comments and I believe the manuscript has been improved. I have only a few minor comments.
- I believe there is a typo in Table 1. In the manuscript, it says the mean for self-control for the unrelated disorder is 4.23, but in the table, it says 3.23.
-
For both studies, please specify the amount that participants were paid for their participation.
-
I was surprised to see the addition of Bayesian analyses in the manuscript file. Did another reviewer suggest this? I can only see my own comments and your responses, so I am a little confused. Were these analyses pre-registered, and if not, please provide justification for adding them.
Author Response
Reviewer 2
- I believe there is a typo in Table 1. In the manuscript, it says the mean for self-control for the unrelated disorder is 4.23, but in the table, it says 3.23.
Thank you for your attention to detail. This has been fixed in the table. The correct value is 4.23.
- For both studies, please specify the amount that participants were paid for their participation.
Each study took approximately 20 minutes and participants were compensated $1.00. This information has been added to both studies in the method sections (pages 5 and 8).
- I was surprised to see the addition of Bayesian analyses in the manuscript file. Did another reviewer suggest this? I can only see my own comments and your responses, so I am a little confused. Were these analyses pre-registered, and if not, please provide justification for adding them.
Yes, these analyses were requested by another reviewer in order to increase the informativeness of our results, specifically by providing evidence for acceptance of null versus hypothesized effects. We make clear that this is the goal of these hypotheses, for example, by stating, “We also employed Bayesian independent t-tests to better assess the extent to which we can accept versus reject the null hypotheses (H0)—that there are not difference in perceptions of self-control or morality of a higher-weight individual based on whether she has a weight-related disorder that causes weight gain or a weight-unrelated disorder—relative to predicted differences by condition (H1)” in Study 1 results (p. 6).
